# AUTOMATED ZONAL LEVEL IMPLANT LOOSENING DETECTION FROM HIP X-RAY USING A MULTI-STAGED APPROACH

## ABSTRACT

Hip arthroplasty is a surgical procedure that involves the replacement of a patient's hip joint with a prosthetic implant. While these implants are initially effective, they may eventually fail and necessitate revision surgery. It is important to identify the 3 Charnley and 7 Gruen zones around the implant and then identify the zone-wise radiolucency which indicates loosening for effective pre and post-operative planning. Despite the importance of zones, there is a lack of automation attempts in this field. In this work, we have proposed a 3-stage algorithm that detects the sanity of the image for diagnosis, segments into the zones, and then identifies radiolucency within the zones. We have demonstrated a $94\%$ accuracy for Fit/Not Fit segregation, a $0.95$ dice score for our zonal segmentation, and a $98\%$ overall loosening accuracy. Obtaining an average dice score of $0.92$ in the segmentation of zones and $0.93$ accuracy on loosening detection on a blind dataset indicates the robustness of the proposed algorithm. This work will contribute to the development of more efficient and accurate models to detect implant loosening.

## 1 INTRODUCTION

Total joint replacement (TJR) is a surgical procedure in which parts of an arthritic or damaged joints are removed and replaced with a metal, plastic, or ceramic device called a prosthesis. The prosthesis is designed to replicate the movement of a normal healthy joint. TJR has been around for more than five decades now. There have been significant improvements in materials, design, and technique making these surgical procedures more dependable and reliable. However, as the number of cases grew problems also emerged which needed these prostheses to be revised due to wear and aseptic loosening, sepsis or infection, and periprosthetic fractures as well as dislocations. Aseptic loosening refers to the prosthesis loosening out without any evidence of infection. This is a gradual process, may take several years to manifest, and often be asymptomatic. It may occur as a result of undersized or poor surgical technique or as a consequence of an inflammatory reaction to wear particles generated from polyethylene liners of the joint. This is one of the leading causes of revision and depends on careful analysis of serial x-rays of the implant coupled with clinical assessment of symptoms such as pain or limp. Analysis of an anteroposterior radiograph of the hip can indicate the presence of radiolucent areas around the implant as shown in Fig 1. Progressive radiolucencies or osteolysis suggest that the prosthesis is loose, and also the implant's positional variations about the bone are symptomatic of early loosening.

The widely used clinical protocol for assessing the implant health of femoral prosthesis as described by Gruen and Charnley is by dividing the femoral region into 7 Gruen zones, and the cup region into 3 Charnley zones. This is shown in Fig 2. These zones help in identifying the extent of radiolucency and the severity of loosening.

Loose implants need to be revised. The accurate diagnosis and effective intervention in orthopedic surgery heavily relies on the expertise of experienced radiologists and orthopedic surgeons. In regions with limited access to orthopedic surgeons, particularly in underdeveloped and developing nations, the healthcare system faces significant strain. Identifying the areas of implant loosening and formulating a revision plan is a time-consuming task for experts when performed manually.

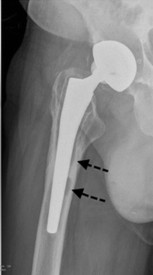

Figure 1: Presence of radiolucency areas (black arrow) indicating loosening of the implant Apostu et al. (2018)

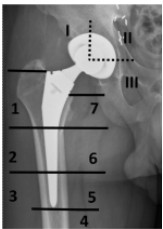

Figure 2: Standard template for radiographic assessment of periprosthetic lucency, with 3 acetabular Charnley zones (I–III) and 7 femoral Gruen zones (1–7) Vanrusselt et al. (2015)

While the ultimate decision rests with the medical professionals, providing assistance in identifying radiolucency, subsidence, or loosening can expedite decision-making and enhance surgical planning for patients Christian et al. (2022). An automated tool can support these professionals in reaching decisions more efficiently and consistently, as it does not have inter-observer variability.

## 2 LITERATURE REVIEW

The challenge of identifying various anomalies in implants has been addressed by a few studies. There are few works reported on implant loosening detection. Rahman *et al.* created the aseptic hip loosening Kaggle dataset comprising 206 images for detecting hip arthroplasty loosening Rahman et al. (2022). They developed an object detection model using YOLO V5 to localize implants. They reported an accuracy of $94\%$ for loosening detection using DenseNet. Alireza *et al.* conducted a study focusing on hip loosening using DenseNet201 on a private dataset comprising 236 cementless total hip replacement (THR) images from a cohort of 15277 THR patients. The model yielded an area under the curve (AUC) of $84\%$, a specificity of $75.0\%$, and a sensitivity of $91.6\%$ Alireza et al. (2019). Lau *et al.* implemented the Xception neural network for 2-class classification in the categories of "loose" or "control" and obtained an accuracy of $96.3\%$ Lawrence et al. (2022). Kim *et al.* Kim et al. (2023) investigated the effectiveness of using a deep convolutional neural network (CNN) to identify the loosening of total knee arthroplasty (TKA) implants on plain radiographs using transfer learning. They utilized a pre-trained VGG and experimented with two techniques. In the first technique, they removed the fully connected layer and added a new one to create a new model. The convolutional layer was frozen without training, and only the fully connected layer was trained. In the second technique, a new model was created by adding a fully connected layer and adjusting the range of freezing for the convolutional layer. The second model demonstrated higher accuracy at $97.5\%$. Shah *et al.* Shah et al. (2020) aimed to assess the efficacy of a machine-learning algorithm in diagnosing prosthetic loosening based on preoperative radiographs. The study gathered preoperative radiographs, as well as historical and comorbidity data from 697 patients who underwent revision of total hip or total knee arthroplasty from 2012 to 2018. Convolutional neural network (CNN) models were trained, demonstrating $70\%$ accuracy in identifying prosthetic loosening from radiographs alone. A detailed review paper on artificial intelligence for image analysis in total hip and total knee arthroplasty is given by Gurung *et al.* Gurung et al. (2022).

There is not much work reported on implant zone segmentation. Asma *et al.* focussed on Gruen zone segmentation using a multitask CNN for binary segmentation mapping of the implant and detection of the implant tip point. The authors also utilized the statistical shape model to construct a landmark-based shape model and fit this model to a new image using shape coefficients and pose parameters. The reported dice score for this method was 0.8 Alzaid et al. (2024).

Our research endeavors to provide a thorough health assessment of implants by segmenting them into the Charnley and Gruen zones, rather than simply classifying them as loose or control, as is common in most research papers. The significance of our work lies in three key phases. Firstly, we address the issue of poor image quality, including noise, blurriness, poor exposure, and artifacts, which can lead to misdiagnosis and pose risks during surgical planning. To mitigate this, we have developed a preliminary screening process for X-ray images, allowing surgeons to identify and discard unsuitable images and request a rescan if necessary. Subsequently, we focus on segmenting the implant into 11 zones (7 Gruen, 3 Charnley, and background). Finally, we utilize this zonal information to identify zones with radiolucency, aiding surgeons in making well-informed decisions regarding extent of loosening, affected zones, early signs of loosening, progress monitoring, and the necessity for revision surgery. To the best of our knowledge, we have not encountered any existing work that offers such a comprehensive analysis of the implant.

## 3 OUR CONTRIBUTIONS

The major contributions in this work are summarized as follows.

- Contribution in the creation of a new comprehensive annotations to the existing dataset created by Rahman *et al.* Rahman et al. (2022) which now includes zone-wise segmentation and loosening details of Gruen and Charnley zones and the extent of loosening information. The original annotation dataset had only control and loose image information.

- Proposing an automated approach to check whether the image is fit for diagnosis.

- Proposing a multi-staged deep learning based approach to perform zone-wise segmentation and loosening detection.

- Obtaining good accuracy when blind-tested on a new dataset has proven the robustness and reliability of our proposed network.

### 3.1 DATASET PREPARATION

Upon our thorough review, we have found that there is currently no open-source dataset containing zonal regions (Gruen and Charnley) annotation and zone-wise loosening information available to the best of our knowledge. Acquiring this information would significantly aid in accurately identifying radiolucency regions and improving the development of automated preplanning tools for implant revision surgery. Our study utilized the implant loosening detection dataset created by Rahman *et al.* Rahman et al. (2022), which consists of 206 single hip implant images, comprising 112 loose and 94 control images. These images were meticulously annotated by an orthopedic surgeon using the 'labelme' annotation tool Russell et al. (2008), encompassing various zones, regions, landmark points, and line-based annotations in both JSON and text formats. The annotations cover three Charnley zones 1 to 3, seven Gruen zones 4 to 10, a femoral cup, and a stem, along with landmark points such as the lesser trochanter, greater trochanter, ischium, medullary canal, pubis bone, neck of the femoral stem, shoulder of the femoral stem, center of the femoral cup, acetabular component, and calcar femoral. Furthermore, two lines were included: Line 1 for the lesser trochanter and Line 2 for the middle thirds. An illustrative example of the manual annotation is presented in Fig 3.

In conjunction with the JSON file, an Excel file has been created to indicate whether the image is suitable for diagnosis, identify the presence of radiolucency, categorize the 10 zones (Gruen and Charnley) as loose or control, determine any dislocation or missing cup, and pinpoint any breakage or subsidence, specifying the affected zones and the total number of loose or control zones. Some of these additional information has been utilized in our network for accurate segmentation and classification of zones.

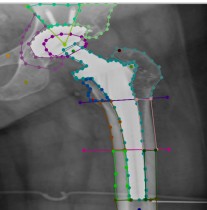

Figure 3: A sample manually annotated image using labelme annotation tool

## 3.2 PROPOSED MULTISTAGED NETWORK

We propose a three-stage network to perform 3 important tasks; image data sanity check, Charnley and Gruen zone region segmentation, and zone-level implant loosening detection.

### 3.2.1 STAGE 1 - FITNESS CHECK OF THE X-RAY IMAGE FOR DIAGNOSIS:

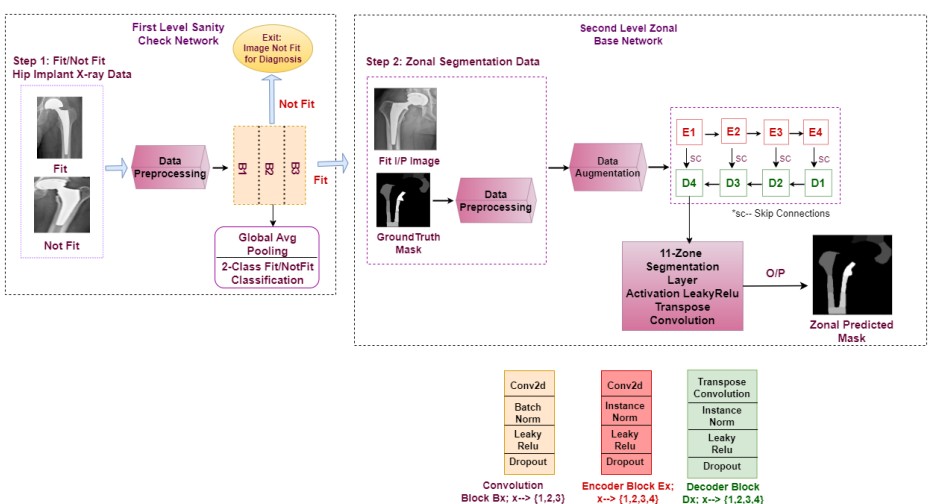

Figure 4: Block diagram for image sanity check and zone segmentation

While the orthopedic surgeon was annotating the dataset it was noted that out of 206 images in the dataset 19 images were deemed not fit for diagnosis. We propose an automated network that can check the fitness of the image for diagnosis. The block diagram for checking this fitness and zonal region segmentation is shown in Fig 4. The network consists of a convolutional neural network (CNN) that is structured into three blocks, each consisting of 2D convolutional blocks, batch normalization, leaky rectified linear unit (ReLU) activation, and dropout layers (0.6). The blocks had 64, 128, and 512 filters, respectively. This was followed by global average pooling and a two-class 'fit' and 'not fit' classification layer. The model was trained using categorical cross-entropy loss and the Adam optimizer with a learning rate set at 0.002, over 100 epochs. Additionally, dynamic learning rate scheduling techniques including time-based decay and a high-factor (0.9) reduction on the plateau were incorporated to optimize training efficiency and performance.

### 3.2.2 STAGE 2 - IMPLANT ZONAL REGION SEGMENTATION:

Identification of 7 Gruen and 3 Charnley zones along with the loosening condition from each zone of the preplanning x-ray image would enable the orthopedic surgeons to better understand the implant condition and also help in planning the surgeries effectively. To achieve this, we converted the annotated JSON file to an XML file and extracted the coordinates of each of the 10 zones. Our primary aim was to identify any radiolucencies present in these specific regions, as radiolucency serves as a critical indicator of loosening and manifests exclusively within these 10 specific areas.

We then created masks of these regions by marking the pixels of each class in a categorical template (0 to background followed by 3 Charnley zones as $1 - 3$ and then followed by 7 Gruen zones from $4 - 10$). During the creation of these masks, we encountered the challenge of overlapping regions in the annotations, which we carefully addressed to ensure separation and non-overlapping markings.

The proposed segmentation network is structured as an encoder-decoder block as shown in Fig 4, wherein the encoder comprises 4 blocks, each consisting of 2D convolution, instance normalization, leaky ReLU activation, and dropout layers. It utilizes 64, 128, 512, and 1024 filters with a $3 \times 3$ kernel size and he_norm as the kernel initializer. The decoder has the same 4 reverse blocks with 2D transpose convolution. The final layer of the network consists of an 11-class segmentation layer with leaky ReLU activation and transpose convolution. This layer serves to segment the input image into 10 distinct zones and a background. Moreover, skip connections are implemented between the encoder and decoder, enhancing the overall performance of the network during the segmentation process. The network employs the Adam optimizer with a learning rate of $2^{-4}$ and a beta value of $0.5$. A batch size of 8 is utilized, and the network is trained for 300 epochs.

The concept of semantic segmentation loss can be categorized into three primary groups: Pixel-level, Region-level, and Boundary-level. The proposed segmentation network adopts a combination approach that integrates elements from the pixel-level and region-level categories to optimize semantic segmentation performance. This approach has significantly enhanced model performance, particularly in the segmentation of 11 challenging classes, including 7 Gruen, 3 Charnley, and the background region, each region with varying shapes and sizes across all images. The strategy of incorporating multiple loss functions aims to strike a balance between pixel-wise precision, overall object segmentation quality, and accuracy. In this approach, the cross-entropy loss (ce) ensures smooth gradients and precise segmentation at each pixel, while the dice loss (dc) prevents local minima and focuses on overall class segmentation by maximizing the alignment between the predicted segmentation and the ground truth mask. The combination of the losses is achieved by summing the cross-entropy loss and the dice loss using an equal weighted contribution of each loss function. The overall equation is defined as follows:

$$\mathbf{L}_{combo} = L_{ce} + L_{dc} \tag{1}$$

where the cross-entropy controls the model's penalization for different target classes in the predicted output. The reduction technique of sum over batch size is used in cross-entropy loss.

$$\mathbf{L}_{dc} = \left(1 - 2 * \frac{num}{den}\right) \tag{2}$$

$$num = Y \cap P \tag{3}$$

$$den = |Y| + |P|; \tag{4}$$

where $Y$ is the ground truth image and $P$ is the predicted image.

### 3.2.3 STAGE 3 - IMPLANT ZONAL LOOSENING DETECTION:

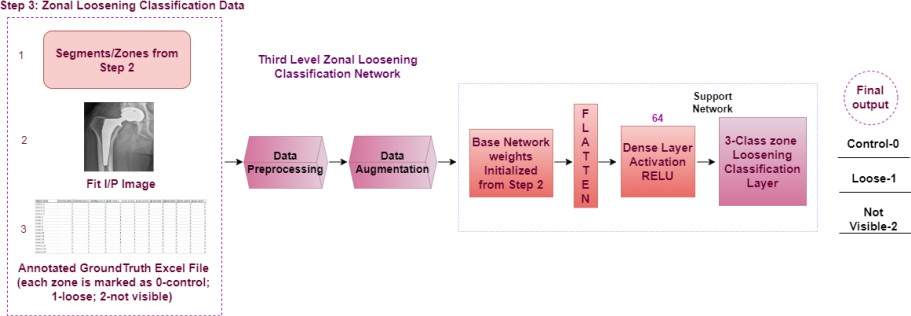

Figure 5: Block diagram for zone-wise loosening detection

The block diagram to detect loosening is given in Fig 5. In this stage, the proposed network is designed to take three inputs: zonal segmentation information from stage 2, the input image, and zonal

loosening information from the created Excel with the help of a domain expert which includes the information on whether the zone is control (0) indicating no signs of radiolucency, loose (1) indicating the presence of radiolucency, or not visible (2) indicating that the zone is not visible for each image. It has to be noted that if any zone is even partially, slightly, or completely invisible, experts recommend a rescan of that X-ray image. The proposed network incorporates a "not visible" class for zones to prevent misjudgment or omissions. So, If any zone within an image is not visible, the system flags it for expert review to facilitate well-informed decisions. When evaluating orthopedic implant integrity, it is crucial to ensure adequate radiolucency visibility around the femoral stem or cup in all 10 zones.

The proposed network comprises two sections: the base network, and a support network. The base network is initialized with the best working weights of the stage 2 network. The small support network consists of a flattening layer, a fully connected dense layer with 64 filters, $ReLU$ activation, and $L2$ kernel regularization, followed by a 3-class classification layer with softmax activation. The hyperparameters such as learning rate, optimizer, and batch size are similar to those in stage 2, and the network ran for 200 epochs. This network provides zone-wise information on the presence of radiolucency implying loosening and also the extent of loosening.

The incorporation of Exponential Logarithmic Loss in this stage, as an extension of the combined loss from the second step, encompasses the exponential logarithmic loss of both cross-entropy and dice losses before their amalgamation. This method effectively addresses the issue of class imbalance, where most areas are categorized as control zones, while a few zones display radiolucency(loose). This method offers the flexibility to regulate the model's focus on easy and hard pixels. The equation is expressed as:

$$L_{Exp\_Log} = L_{Exp\_Log\_dc} + L_{Exp\_Log\_ce} \tag{5}$$

where $L_{Exp\_Log\_dc}$ represents the exponential logarithmic dice loss and $L_{Exp\_Log\_ce}$ signifies the exponential logarithmic cross-entropy loss.

## 4    RESULTS AND DISCUSSION

Out of 206 images in the dataset Rahman et al. (2022), 19 were considered not fit for diagnosis by a participating domain expert while annotating the dataset. The remaining 187 images were split into $70 : 30$ ratios for training and testing the segmentation and loosening algorithms of stage 2 and 3. We have used 130 images for training and 57 images for testing both the proposed segmentation and loosening detection algorithms. The dimension of the original image was $331 \times 331 \times 3$. These images were zero-padded to obtain a size of $336 \times 336 \times 3$ and the pixel data was rescaled to a range of $0 - 1$ to meet the network requirement. To mitigate overfitting and enhance generalization and convergence, on-the-fly data augmentation techniques such as rotation, zooming, brightness adjustment, and horizontal and vertical flipping were implemented due to the limited sample size. We ensured that at least $75\%$ of the original images were retained during training. To ensure repeatability of results we have performed 5-fold cross-validation and the reported results are the average values of this cross-validation.

### 4.1    IMAGE FIT OR NOT FIT RESULTS

The dataset has 19 images that were considered not fit for diagnosis by the domain expert who annotated this dataset. Due to the limited number of samples, data augmentation was applied and the model was successfully trained using an $80 : 20$ ratios train-test split rather than the $70 : 30$ ratios as mentioned earlier. We achieved an accuracy of $94\%$ in classifying images as either fit or not fit for diagnosis. Some of the images flagged as 'not fit' for diagnosis from our proposed method are given in Fig 6. It can be seen that Fig 6a is noisy and inadequate visualization of prosthetic components, while Fig 6b has some textual information in the diagnostic regions and Fig 6c has been poorly exposed.

### 4.2    IMPLANT ZONAL REGION SEGMENTATION AND LOOSENING RESULTS

A notable challenge in the dataset was the variability in the visibility of the 11 zones in the images, with some images showing less than the total number of zones. On close observation with the data

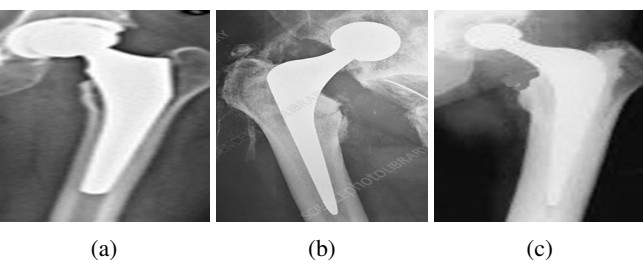

Figure 6: Few images classified as Not fit by our proposed algorithm

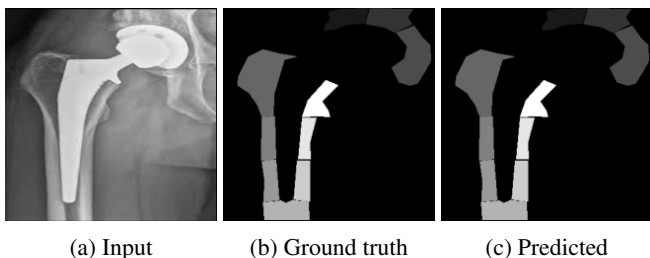

(a) Input        (b) Ground truth        (c) Predicted

Figure 7: Zone-wise segmentation results of the proposed method

it was noted that some images had limited visibility of the three Charnley zones and also Gruen zone region 4. However, our system is adeptly designed to effectively navigate these challenges and deliver accurate zone-wise segmentation for each image. The segmentation accuracy is computed using dice score, which is the common evaluation metric used in computer vision and medical imaging for measuring the similarity or overlap between two sets the ground truth and the predicted images. We found that by using only dice loss we achieved a segmentation accuracy of $87\%$. However, the combination of pixel-level categorical cross-entropy loss function and region-level dice loss significantly improved the performance of zonal segmentation to $95\%$, thereby enhancing the accuracy of each zonal segmentation results as well. The proposed algorithm results are very close to ground truth as can be seen in Fig 7.

The dice score for each zone is given in Table 1. It can be seen that the proposed method has high dice score for all the regions. Our method segments the zones consistently even when they are partially visible in the images.

Table 1: Zone-wise segmentation and loosening classification results

|  | Segmentation | Loosening Classification | | | |
|---|---|---|---|---|---|
|  | Dice score | Precision | Recall | F1-score | Accuracy |
| Charnley Zone 1 | 0.93 | 0.91 | 0.92 | 0.91 | 0.95 |
| Charnley Zone 2 | 0.95 | 0.95 | 0.95 | 0.95 | 0.96 |
| Charnley Zone 3 | 0.93 | 0.89 | 0.94 | 0.91 | 0.95 |
| Gruen Zone 1 | 0.96 | 0.96 | 0.98 | 0.97 | 0.97 |
| Gruen Zone 2 | 0.94 | 0.95 | 0.98 | 0.96 | 0.97 |
| Gruen Zone 3 | 0.94 | 0.94 | 0.98 | 0.96 | 0.97 |
| Gruen Zone 4 | 0.95 | 0.96 | 0.99 | 0.97 | 0.98 |
| Gruen Zone 5 | 0.96 | 0.93 | 0.98 | 0.95 | 0.97 |
| Gruen Zone 6 | 0.95 | 0.94 | 0.97 | 0.95 | 0.96 |
| Gruen Zone 7 | 0.95 | 0.98 | 0.99 | 0.99 | 0.99 |
| Background | 1.0 | - | - | - | - |

The gradient-weighted class activation mapping (GradCam) heatmap of the proposed segmentation network shows the concentration of features around the implant region as shown in Fig 8. Notably, in the feature map, blue reflects the most significant features, yellow denotes moderate significance, and red represents the least significant features. It can be seen that the most significant features are

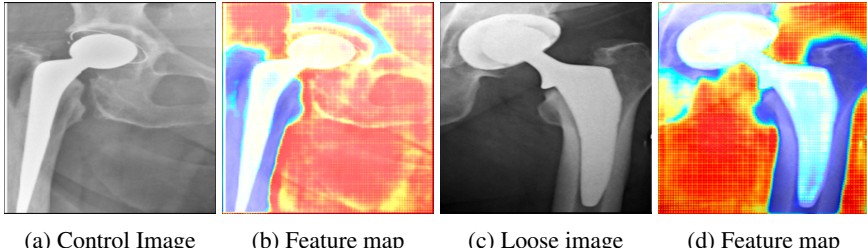

(a) Control Image    (b) Feature map    (c) Loose image    (d) Feature map

Figure 8: GradCam feature map of proposed segmentation method for both control and loose images. The blue reflects the most significant features, yellow denotes moderate significance, and red represents the least significant features.

concentrated near the implant boundaries for both Gruen and Charnley zones suggesting that our network is picking the right features for segmentation.

We compare the performance of our proposed loosening detection classifier using the metrics accuracy, precision, recall, and F1-score. Accuracy is the ratio of correct predictions to all predictions. Precision is the ratio of true positives to predicted positives. Recall is the ratio of true positives to actual positives. F1-score is the harmonic mean between precision and recall. In our final classification, radiolucency in any single zone indicates implant loosening, while the absence of radiolucency in all zones leads to its categorization as a control. This approach facilitates the monitoring of loosening progression and aids in determining whether the intervention should target the head (Charnley zones), the stem (Gruen zones), or both. As a result, it offers a significant practical advantage for clinical applications. The precision, recall, F1-score, and accuracy for each zone in loosening classification are given in Table 1. It can be seen that the proposed has a very high classification score for all the metrics considered in classifying the given image as either control or loose. The confusion matrix for classification is given in Table 2. It can be seen that the proposed has just 1 false negative with no false positives, 23 true positives and 33 true negatives.

Table 2: Classification confusion matrix for loosening detection

|  |  | **Prediction outcome** | |
|  |  | **Control** | **Loose** |
| **Actual value** | **Control** | True Positive (23) | False Negative (1) |
|  | **Loose** | False Positive (0) | True Negative (33) |

The Gradcam heatmap of the proposed loosening classification network is shown in Fig 9. It can be seen that the features are concentrated on locations where there might be potential radiolucency. As mentioned earlier, blue reflects the most significant features, yellow denotes moderate significance, and red represents the least significant features. In our proposed loosening detection algorithm of stage 3, the regions with moderate radiolucency are highlighted in yellow, demonstrating the network's ability to differentiate areas with higher radiolucency from those with less or no radiolucency in an image thereby illustrating visually the effectiveness of our system.

Table 3 compares the performance of our proposed method with other methods reported in the literature on the same dataset. The proposed exceeds other methods with high 98% accuracy, 99% precision, and 98% recall and F1-score indicating the robustness of the proposed approach.

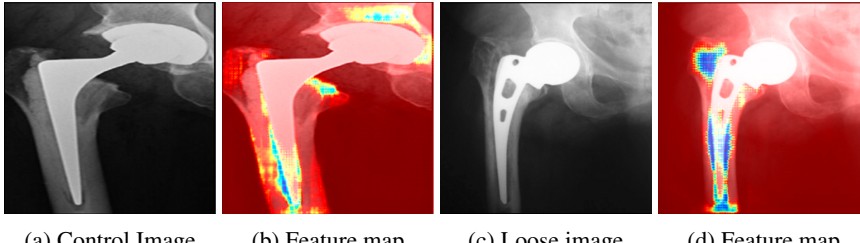

| (a) Control Image | (b) Feature map | (c) Loose image | (d) Feature map |

Figure 9: GradCam feature map of proposed loosening detection method for both control and loose images. The blue reflects the most significant features, yellow denotes moderate significance, and red represents the least significant features.

Table 3: Comparison of performance measure of loosening with other reported methods on the same dataset.

| Methods | Accuracy | Precision | Recall | F1-score |
|---|---|---|---|---|
| Densenet201 Rahman et al. (2022) | 94.66 | 94.66 | 94.66 | 94.66 |
| Random Forest Rahman et al. (2022) | 96.11 | 96.42 | 96.42 | 96.42 |
| Xception Alireza et al. (2019) | 77 | 83 | 75 | 91 |
| Dense Net Lawrence et al. (2022) | 95 | 94 | 96 | - |
| Proposed | **98** | **99** | **98** | **98** |

## 4.3 BLIND TESTING

The proposed approach is tested on anonymized clinical data obtained from an orthopedic surgeon. This dataset contains 38 THR images of diverse dimension and configurations like changes in lighting conditions, machine settings, patient positions, film processing, and radiographic techniques capturing the entire hip area. Preprocessing on these images involved cropping to isolate the right and left implant regions, along with resizing and normalization to align with our proposed model. It has to be noted that none of these images were utilized for training the model. Some of the results of the blind testing are given in Fig 10.

Table 4 shows the performance of our method on blind tested dataset. During the evaluation of zone-wise segmentation phase, the proposed methodology achieved an average dice score of 0.92. The average blind tested accuracy on loosening detection is 0.93. Obtaining a high zone-wise precision, recall, F1-score, and accuracy as shown in Table 4 on a new unseen dataset indicates the repeatability of our method.

Table 4: Zone-wise segmentation and loosening classification results on blind testing dataset

| | Segmentation | Loosening classification | | | |
|---|---|---|---|---|---|
| | Dice score | Precision | Recall | F1-score | Accuracy |
| Charnley Zone 1 | 0.88 | 0.83 | 0.97 | 0.88 | 0.95 |
| Charnley Zone 2 | 0.90 | 0.94 | 0.86 | 0.90 | 0.84 |
| Charnley Zone 3 | 0.89 | 0.83 | 0.85 | 0.81 | 0.82 |
| Gruen Zone 1 | 0.92 | 1.00 | 1.00 | 1.00 | 1.00 |
| Gruen Zone 2 | 0.92 | 0.93 | 0.98 | 0.95 | 0.97 |
| Gruen Zone 3 | 0.91 | 0.88 | 0.97 | 0.91 | 0.95 |
| Gruen Zone 4 | 0.93 | 0.98 | 0.87 | 0.91 | 0.95 |
| Gruen Zone 5 | 0.92 | 0.98 | 0.93 | 0.95 | 0.97 |
| Gruen Zone 6 | 0.95 | 0.94 | 0.98 | 0.96 | 0.97 |
| Gruen Zone 7 | 0.91 | 0.90 | 0.87 | 0.89 | 0.92 |
| Background | 1.0 | - | - | - | - |

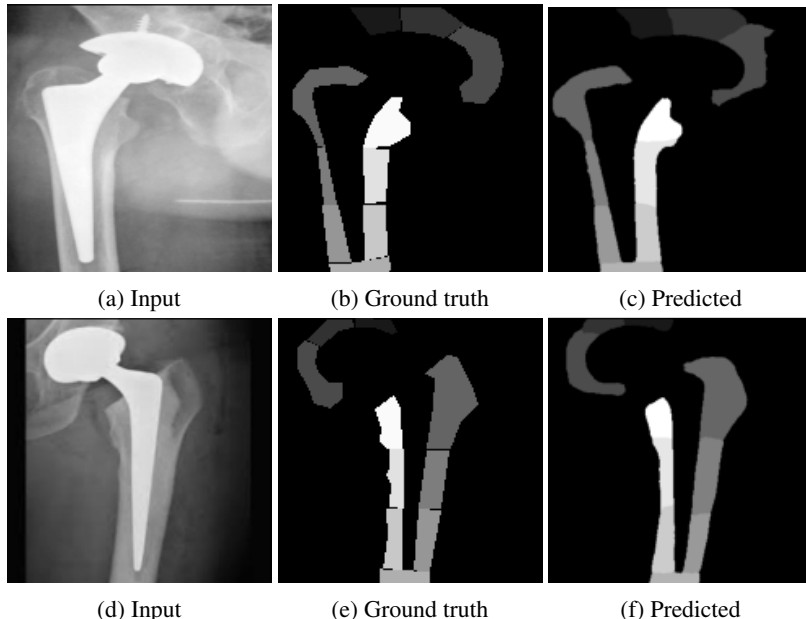

(a) Input (b) Ground truth (c) Predicted

(d) Input (e) Ground truth (f) Predicted

Figure 10: Blind testing segmentation results. None of these images were used for training the network.

## 5 CONCLUSION

In this study, a comprehensive analysis of implants has been proposed, encompassing the assessment of image quality, segmentation of the Gruen and Charnley zones, and the analysis of radiolucency within these zones to determine the extent of implant loosening. This information is vital for both planning implant revision surgeries and monitoring the overall implant health. Collaborating with an orthopedic expert, we have developed an annotated dataset for zone segmentation and an accompanying Excel sheet containing zonal loosening information, as no such open-source dataset was available. Our segmentation process achieved an average dice score of 95% for segmenting into the standard zones, and an accuracy of 98% in identifying images indicative of loosening. Additionally, we flagged images with artifacts, blurriness, or inadequate zone visibility. The proposed work also assists the surgeons in determining cases that require immediate attention and also in identifying any early signs of loosening, and monitoring cases that require careful follow-up. Our future work aims to incorporate patients' imaging history to identify any dislocations or implant sinking, along with detecting anomalies such as fractures, implant subsidence, and breakage. This comprehensive approach will facilitate the development of a complete implant health assessment suite.

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
