# OpenReview forum: "Automated Zonal level implant loosening detection from Hip X-ray using a multi-staged approach"
_ICLR.cc/2025/Conference — Submitted to ICLR 2025_

### Official Review · Reviewer_BamT · 2024-10-21

**Soundness:** 2
**Presentation:** 2
**Contribution:** 1
**Rating:** 3
**Confidence:** 4

**Summary:**

This paper addresses analysis of radiographs of hip protheses. Specifically, the authors propose a system based on deep learning to perform three analysis tasks automatically: (i) assessment of image suitability, (ii) segmentation of ten zones that are used in clinical practice, and (iii) detection of loosening. Evaluation is reported using a dataset of 206 images to which new annotations were added by a radiologist.

**Strengths:**

The zone segmentation and loosening classification results seem to be good (using standard methods on a relatively small dataset).

**Weaknesses:**

The dataset has only 19 “not fit” examples which is too few to train and test the stage 1 classifier. The accuracy reported for this binary classification task is 94% which is not significantly different from chance level which is 91%.

The dataset was annotated by one orthopaedic surgeon and no indication of annotation reliability was provided. It would be preferable to have multiple annotations of at least a subset to give some idea of inter/intra annotator variability. I imagine there is non-negligible variability in the boundaries between adjacent zones, for example.

The methods used for image segmentation and classification are fairly standard. It would be good to compare with some pre-trained off-the-shelf architectures. The stage 2 network sounds like it is a U-Net but it is not called that in the paper nor is any prior work on U-Nets cited. The loss employed is also quite widely used but again prior work is not cited to make this clear.

**Questions:**

The use of cross-validation across the 3 tasks on this small dataset needs further clarification, to reassure that there has not been data leakage.
Section 4 states the data-split was 70% training and 30% testing which seems inconsistent with the statement that the results are averaged over splits in a 5-fold cross-validation. Can this be clarified?

Is the fully annotated dataset available to other researchers? Is the code?

How was left-right mirroring handled (see e.g. Figure 6)?

It would be useful to clarify how overlapping regions in the annotations were “carefully addressed to ensure separation and non-overlapping markings”. Should adjacent zones not be contiguous?

The loosening classification result in Table 2 seems surprisingly good given that there are presumably some boundary cases which radiologists might disagree on. It would be useful to see such boundary cases given as examples and discussed. Similarly, it would be good to see error examples and discussion for the segmentation problem.

---

### Official Review · Reviewer_azjs · 2024-11-01

**Soundness:** 3
**Presentation:** 3
**Contribution:** 2
**Rating:** 5
**Confidence:** 3

**Summary:**

The paper proposes a 3-stage deep learning algorithm for detection of implant loosening after hip replacement surgery. The method involves fitness check of images, region segmentation and loosening detection. Also, Gruen and Charnley zones annotation and zone-wise loosening information are built on a open-source dataset. The experimental results demonstrate high accuracy of the proposed method.

**Strengths:**

1. The 3-stage algorithm provide a thorough assessment of implants for the Charnley and Gruen zones, comparing with the common classification of other papers.
2. The experimental results provide evidence of the superiority of this method over other reported methods , which supports its effectiveness.
3. The overall writing is clear and well-structured.

**Weaknesses:**

1. The originality of the paper, which proposes a method to segment regions and classify loosening, may not be seen as highly innovative. The techniques of image segmentation and classification have been well-explored in deep learning research.
2. The limited availability of data for training and testing the algorithm may limit its effectiveness in real-world scenarios. The blind testing contains only 38 images, which is not enough to prove its robustness.

**Questions:**

1. I recommend that the paper includes a more extensive dataset to robustly demonstrate the effectiveness of the proposed method.
2. The explanations of Figure 3 could be more detailed for further understanding of stage 2 and stage 3.
3. Consider provide clearer explanations of loss function formulas  for better precision in mathematical representation.

---

### Official Review · Reviewer_daX7 · 2024-11-02

**Soundness:** 2
**Presentation:** 2
**Contribution:** 2
**Rating:** 1
**Confidence:** 5

**Summary:**

The author present a mult-step segmentation approach for a medical image segmentation task. They contribute to the anntoation of a public dataset and report favorable performances in their application

**Strengths:**

* The authors adddress an interesting application
* Reported performances are good
* The contribution to a public resource / dataset is appreciated

**Weaknesses:**

* The study focuses on a very narrow application and the solution proposed is highly specialized on this application
* It is not quite clear what type of segmentation algorithm is used, important technical details are missing (also elsewhere)

**Questions:**

* Can you relate this work to prior publications in ICLR or closely related confrences? (Right now there are mostly clinical application referenced.)  It would make this study more accesible to the ICLR audience. And it would help you to sharpen the presentation of the contributions that are relevant to the ICLR audience.

---

### Official Review · Reviewer_kYcN · 2024-11-04

**Soundness:** 1
**Presentation:** 1
**Contribution:** 1
**Rating:** 1
**Confidence:** 5

**Summary:**

The authors have proposed proposed a 3-stage algorithm that detects the sanity of the image for diagnosis, segments into the zones, and then identifies radiolucency within the zones.

**Strengths:**

1. It is a novel application of existing deep learning techniques to the field of orthopaedics.
2. Overall empirical scores seem to be quite high, which indicates a good potential of clinical translation.
3. GradCAM maps (issue discussed below) seem to provide good qualitative confidence in terms of importance regions developed by the network.

**Weaknesses:**

1. The primary results stem from high quality annotations rather than theoretical/architectural advancements. Maybe more suited to a different conference. What could be interesting is a multi-task model as the it can be seen a common backbone can be used from all and the there is a definite inter task benefit which could be presented as a custom loss (for a baseline comparison at the least).

2. The overall outline is not very clear to me in terms of the actual task at the final stage. In Stage 3, to my understanding it is classification, but the section discusses of a segmentation loss? A general figure covering the entire pipeline is highly suggested, which combines Figure 4 and 5 along with an additional figure for stage 2.

**Questions:**

1. Figure 8 shows the GradCam of the segmentation method. How exactly is this done? The baseline GradCAM is only suitable for classification models.
2. "We ensured that at least 75% of the original images were retained during training" -- What is the rationale behind this? Does this mean the test set was augmented?
3. Panoptic Quality may be a better score to analyze the results instead of dice score:
- Given there are majority classes even based on size here.
- It will be one metric representative of the entire image instead of a class by class analysis (which is definitely good to keep in as well.)

**Details Of Ethics Concerns:**

Not sure if the dataset is open-source, if so based on what compliance?

---

### Meta-Review · Area_Chair_7XMS · 2024-12-19

**Metareview:**

The present manuscript provides segmentation and pre-/post-processing support for intra-operative imaging in total hip arthroplasty, annotating femoral and acetabular/pelvic component regions for the prediction of implant loosening.

While all reviewers agree that this work is on a topic worthy of study and that results are promising (albeit preliminary/limited to small dataset), it is also the view of all reviewers that this work is not correctly placed for the ICLR community. While "applications papers" are regularly accepted at ICLR, in general those manuscripts offer methodological advancements that are driven by the application problem. As noted by two reviewers, this manuscript does not offer those contributions.

I strongly recommend the authors consider a venue closer to medical imaging such as MIDL, MICCAI, or IPCAI, and/or surgery, such as the numerous journal venues (e.g., IJCARS); I find that this work has merit, but I agree that the merits of this work are not appreciable by the ICLR community, or even the small subset of medical imaging people that attend ICLR.

**Additional Comments On Reviewer Discussion:**

There are numerous improvements to methodology that are suggested by the reviewers that are also relevant to improvement, but not to the rational for rejection. Reviewers `azjs` and `BamT` both suggest dataset improvements; while data collection is generally expensive, given the number of THA procedures performed per attending physician (or equivalent), it might be feasible to expand the dataset, even considering that it might be enriched for loosening factors from the general procedure-receiving population.

Reviewer `BamT` also notes that inter-rater reliability is not measured. While this may not be feasible due to physician expense, if annotation is not a bottleneck it would benefit the study to show that these results are reproducible between physicians, and that we haven't overfit to the idiosyncrasies of either the dataset (first point, dataset size) or the annotator (second point, inter-rater reliability).

---

### Decision · Program_Chairs · 2025-01-22

Reject